# Adversarial Attacks on Online Learning to Rank with Click Feedback

**Jinhang Zuo**[1,2]    **Zhiyao Zhang**[3]    **Zhiyong Wang**[4]    **Shuai Li**[5][*]
**Mohammad Hajiesmaili**[1]    **Adam Wierman**[2]
[1]University of Massachusetts Amherst  [2]California Institute of Technology
[3]Southeast University  [4]The Chinese University of Hong Kong  [5]Shanghai Jiao Tong University
{jhzuo,hajiesmaili}@cs.umass.edu  muirheadzhang@gmail.com
zywang21@cse.cuhk.edu.hk  shuaili8@sjtu.edu.cn  adamw@caltech.edu

## Abstract

Online learning to rank (OLTR) is a sequential decision-making problem where a learning agent selects an ordered list of items and receives feedback through user clicks. Although potential attacks against OLTR algorithms may cause serious losses in real-world applications, there is limited knowledge about adversarial attacks on OLTR. This paper studies attack strategies against multiple variants of OLTR. Our first result provides an attack strategy against the UCB algorithm on classical stochastic bandits with binary feedback, which solves the key issues caused by bounded and discrete feedback that previous works cannot handle. Building on this result, we design attack algorithms against UCB-based OLTR algorithms in position-based and cascade models. Finally, we propose a general attack strategy against any algorithm under the general click model. Each attack algorithm manipulates the learning agent into choosing the target attack item $T - o(T)$ times, incurring a cumulative cost of $o(T)$. Experiments on synthetic and real data further validate the effectiveness of our proposed attack algorithms.

## 1 Introduction

Online learning to rank (OLTR) has been extensively studied [1, 2, 3, 4, 5] as a sequential decision-making problem where, in each round, a learning agent presents a list of items to users and receives implicit feedback from user interactions. One of the most common forms of feedback considered in literature is in the form of user clicks [1, 4, 5]. OLTR with such click feedback can lead to major improvements over traditional supervised learning to rank methods [6, 7, 8]. However, there is a security concern that user-generated click feedback might be generated by malicious users with the goal of manipulating the learning agent. Understanding the vulnerability of OLTR under adversarial attacks plays an essential role in developing effective defense mechanisms for trustworthy OLTR.

There has been a surge of interest in adversarial attacks on multi-armed bandits [9, 10, 11]. For example, [9] shows that, for stochastic bandits, it is possible to manipulate the bandit algorithm into pulling a target arm very often with sublinear cumulative cost. Though it generally follows the bandit formulation, it differs from stochastic bandits in the action space and feedback model. More specifically, OLTR chooses a list of $K$ out of $L$ arms, instead of just one arm, to play in each round; the realized rewards of the chosen arms are usually censored by a click model, e.g., position-based [4] or cascade model [5]. Thus, it is nontrivial to design efficient adversarial attacks on the censored feedback of the chosen arms for OLTR.

Moreover, previous works [9, 10] can only handle unbounded and continuous reward feedback, while the (binary) click feedback of OLTR is bounded and discrete. Such binary feedback brings new

---

[*]Corresponding author.

37th Conference on Neural Information Processing Systems (NeurIPS 2023).

Table 1: Summary of the settings and proposed attack strategies[†]

| Setting | Attack against | $N_L(T)$ | $\lim C(T)/\log T$ |
|---|---|---|---|
| $L$-armed bandits | $UCB$ [12] | $T - O\left((L-1)\left(\frac{1}{\Delta_0^2}\log T\right)\right)$ | $O\left(\sum_{a<L}\frac{\Delta_a+\Delta_0}{\Delta_0^2}\right)$ |
| Position-based model | $PBM\text{-}UCB$ [4] | $T - O\left((L-K)\left(\frac{1+\epsilon}{\kappa_K^2\Delta_0^2}\log T\right)\right)$ | $O\left(\sum_{a<L}\frac{(1+\epsilon)(\Delta_a+\Delta_0)}{\kappa_M^2\Delta_0^2}\right)$ |
| Cascade model | $CascadeUCB$ [5] | $T - O\left((L-K)\left(\frac{1}{p^*\Delta_0^2}\log T\right)\right)$ | $O\left(\sum_{a<L}\frac{\Delta_a+\Delta_0}{\Delta_0^2}\right)$ |
| General model | $Arbitrary$ | $T - O(\log T)$ | $O\left(\sum_{a<L}(\Delta_a + 4\beta(1))\right)$ |

[†] $\Delta_0$: parameter of the attack algorithm; $\Delta_a$: mean gap between arm $a$ and $L$; $\beta$: a decreasing function in Section 3.2

challenges to the attack design against OLTR algorithms. Since the post-attack feedback must also be binary, this scenario introduces a new problem of deciding whether to attack when the output attack value from previous algorithms is between 0 and 1: a simple rounding up might be costly, while skipping the attack may lead to undesired non-target arm pulls. Furthermore, the attack value computed by previous attack algorithms can be larger than 1, which is higher than the maximum attack value in the click model. In other words, in the bounded and discrete model, it is impossible to always find a feasible attack value to ensure the required conditions for their theoretical guarantees.

**Contributions.** In this paper, we propose the first study of adversarial attacks on OLTR with click feedback, aiming to overcome the new issues raised by the OLTR click model as well as the binary feedback. Table 1 summarizes our proposed attack algorithms with their theoretical guarantees. Since the click feedback itself complicates the attack design, we first consider adversarial attacks on stochastic $L$-armed bandits with Bernoulli rewards where the feedback from the chosen arm is always binary. We propose an attack algorithm that can mislead the well-known UCB algorithm [12] to pull the target arm $L$ for $N_L(T)$ times in $T$ rounds, with a cumulative cost $C(T)$ in the order of $\log T$ asymptotically. Based on this approach, we study the two most popular click models of OLTR, position-based model [4] and cascade model [5], and propose attack strategies against UCB-type OLTR algorithms. Their cumulative costs depend on $\kappa_K$ and $p^*$, which are instance-dependent parameters of the position-based and cascade models, respectively. Lastly, we introduce the threat model for OLTR with a general click model and design an attack algorithm that can misguide arbitrary OLTR algorithms using click feedback. Our technical contributions are summarized as follows.

1. We propose the new idea of *conservative estimation* of the target arm for attacking UCB in stochastic bandits with binary feedback, which resolves the issues caused by the bounded and discrete requirements of the post-attack feedback. This approach is also the backbone of other attack strategies in more complicated OLTR settings.

2. The PBM-UCB algorithm uses *position bias-corrected counters* rather than simple click counters to compute UCB indices; we provide the theoretical analysis of our attack algorithm against PBM-UCB by carefully treating these unusual counters.

3. The *partial feedback* of OLTR with the cascade click model brings a new challenge to the attack design. We provide a new *regret-based analysis* of our attack algorithm against CascadeUCB without suffering from the partial feedback issue.

4. We devise a general attack strategy based on a new *probabilistic attack design*. It can successfully attack arbitrary OLTR algorithms without knowing the details of the algorithm.

We also conduct experiments on both synthetic and real-world data to evaluate our proposed attack strategies. Experimental results show that they can effectively attack the corresponding OLTR algorithms with less cost compared to other baselines. Due to space constraints, proofs and empirical results are included in the appendix.

**Related Work**. Online learning to rank with different feedback models has attracted much attention in recent years. Though there are other types of feedback models such as top-$K$ feedback [13], click feedback has been widely used in literature. [4, 14, 15] consider the position-based model (PBM), where each position of the item list has an examination probability known or unknown by the learning agent. The cascade model in [5, 16, 17, 18, 19, 20] considers the setting where the user would check the recommended items sequentially and stop at the first clicked one; all items after the clicked item will not be examined. The dependent click model (DCM) is a generalization of the cascade model where the user may click on multiple items [21, 22]. There are also general click models [1, 2, 3]

that can cover some of the previous click models. In this paper, we mainly focus on attack design on PBM and cascade models; both of them adopt bounded and discrete click feedback thus require a new design other than previous works like [9] that can only handle unbounded and continuous feedback.

Adversarial attacks on different types of multi-armed bandit problems have been studied recently [9, 10, 23, 24, 25]. [9] proposes the first study of adversarial attacks on the classical stochastic bandit problem. It designs effective attack strategies against the $\epsilon$-Greedy and UCB algorithms. [10] extends it to a more general setting, where the algorithm of the learning agent can be unknown. [23] studies adversarial attacks on linear contextual bandits where the adversarial modifications can be added to either rewards or contexts. To the best of our knowledge, we are the first to study adversarial attacks on OLTR where, in addition to the challenges raised by click feedback, the combinatorial action space and the censored feedback of OLTR make it non-trivial to design efficient attack strategies.

## 2 Preliminaries

In this section, we briefly discuss the three problem settings we consider in this paper.

**Stochastic $L$-armed bandit**. We consider an arm set $[L] = \{1, 2, \cdots, L\}$, with $\mu_i$ as the expected reward of arm $i$. Without loss of generality, we assume $\mu_1 \geq \mu_2 \geq \cdots \geq \mu_L$. In round $t$, the player chooses an arm $a_t \in [L]$ to play and receives a reward $r_t^0$ as feedback. In the click feedback setting, the realized reward $r_t^0 \in \{0, 1\}$ of arm $a_t$ is sampled from a Bernoulli distribution with expectation $\mu_{a_t}$. The player aims to find an optimal policy to maximize the long-term cumulative reward.

**OLTR with position-based model [4]**. This setting also considers the item (arm) set $[L]$ where $\mu_i$ represents the click probability of item $i$. However, in OLTR with the position-based model (PBM), in each round $t$, the player chooses an ordered list of $K$ items, $\boldsymbol{a}_t = (a_{1,t}, \cdots, a_{K,t})$, with known examination probability $\kappa_k$ for the $k$-th position in the list (assuming $\kappa_1 \geq \cdots \geq \kappa_K$). The player then observes the click feedback of the chosen list from the user, denoted as $\boldsymbol{r}_t^0 = (r_{1,t}^0, \cdots, r_{K,t}^0) \in \{0, 1\}^K$, where $r_{i,t}^0$ is sampled from a Bernoulli distribution with expectation $\kappa_i \mu_{a_{i,t}}$. The reward obtained by the player is the sum of the clicks, i.e., $\sum_{k=1}^K r_{k,t}^0$. The goal of the player is to find an optimal policy that can maximize the long-term cumulative user clicks.

**OLTR with cascade model [5]**. Here, we consider the same item set $[L]$ as that in the PBM model. For OLTR with cascade model, in each round $t$, the player chooses an ordered list of $K$ items, $\boldsymbol{a}_t = (a_{1,t}, \cdots, a_{K,t})$, for the user. The user then checks the list from $a_{1,t}$ to $a_{K,t}$, with probability $\mu_{a_{k,t}}$ to click the $k$-th item. She immediately stops at the first clicked item, and returns the click result back to the player. We denote the position of the first clicked item as $\mathbf{C}_t$ ($\mathbf{C}_t = \infty$ if no item was clicked). The click feedback of the player is $\boldsymbol{r}_t^0 = (r_{1,t}^0, \cdots, r_{K,t}^0) \in \{0, 1\}^K$, where only $r_{\mathbf{C}_t, t}^0 = 1$ and $r_{k,t}^0 = 0$ for all $k \neq \mathbf{C}_t$. The reward obtained by the player is again the sum of user clicks $\sum_{k=1}^K r_{k,t}^0$, but in the cascade model, it is at most 1. The goal of the player is also to find an optimal policy that can maximize the long-term cumulative user clicks.

## 3 Attacks on Stochastic $L$-armed Bandits with Binary Feedback

As mentioned in the introduction, one main challenge of attacking OLTR algorithms comes from the binary click feedback: such binary feedback limits the possible actions of the attacker since they need to ensure the post-attack reward feedback is still valid (binary). This is a common issue for all OLTR with click feedback. Hence, in this section, we focus on adversarial attacks on the $L$-armed bandit problem with binary feedback. We propose an attack algorithm against the UCB algorithm, which is the backbone of the attack strategies for more complicated OLTR settings.

### 3.1 Threat Model

We first introduce the threat model for the $L$-armed bandit problem with binary feedback. In each round $t$, the player chooses an arm $a_t \in [L]$ to play. The environment generates the pre-attack reward feedback $r_t^0 \in \{0, 1\}$ based on a Bernoulli distribution with mean $\mu_{a_t}$. The attacker then observes $a_t, r_t^0$, and decides the post-attack feedback $r_t \in \{0, 1\}$. The player only receives $r_t$ as the feedback and uses that to decide the next arm to pull, $a_{t+1}$, for round $t + 1$. Without loss of generality, we

---

**Algorithm 1** Attacks against the UCB algorithm on stochastic bandits with binary feedback

---
1: **Initialization**: $h_a(0) = 1$ for all $a \in [L]$
2: **for** $t = 1, 2, 3, \ldots$ **do**
3:     Observe $a_t, r_t^0$
4:     **if** $a_t \neq L$ **then**
5:         Calculate $\gamma_t, \tilde{\gamma}_t$ according to Eqs. (2) and (3)
6:         **if** $\gamma_t \leq r_t^0$ **then**
7:             $\alpha_t = \lceil \gamma_t \rceil$, $h_{a_t}(t) = t$
8:         **else**
9:             $\alpha_t = \lceil \tilde{\gamma}_t \rceil$, $h_{a_t}(t) = h_{a_t}(t-1)$
10:         **end if**
11:     **end if**
12:     Return $r_t = r_t^0 - \alpha_t$; update $h_a(t) = h_a(t-1)$ for all $a \neq a_t$
13: **end for**

---

assume arm $L$ is a sub-optimal *target* arm. The attacker's goal is to misguide the player to pull the target arm $L$ very often while using small attack costs. Let $N_i(t)$ denote the number of pulls of arm $i$ up to round $t$. We say the attack is successful after $T$ rounds if $N_L(T) = T - o(T)$ in expectation or with high probability, while the cumulative attack cost $C(T) = \sum_{t=1}^{T} |r_t - r_t^0| = o(T)$.

### 3.2 Attack Algorithm against UCB

For a better illustration of the attack strategy, we first define the following auxiliary notations. Let $\tau_i(t) := \{s : s \leq t, a_s = i\}$ be the set of rounds up to $t$ when arm $i$ is played. We denote the pre-attack average reward of arm $i$ up to round $t$ by $\hat{\mu}_i^0(t) := N_i(t)^{-1} \sum_{s \in \tau_i(t)} r_s^0$. Lastly, let $\hat{\mu}_i(t) := N_i(t)^{-1} \sum_{s \in \tau_i(t)} r_s$ be the post-attack average reward of arm $i$ up to round $t$.

As in [9], we consider attacks against the $(\alpha, \psi)$-UCB algorithm from [12], where $\alpha = 4.5$ and $\psi : \lambda \mapsto \lambda^2/8$ since Bernoulli random variables are $1/4$-sub-Gaussian. The original attack algorithm in [9] calculates an attack value $\alpha_t$ for round $t$ such that

$$\hat{\mu}_{a_t}(t) \leq \hat{\mu}_L(t-1) - 2\beta(N_L(t-1)) - \Delta_0, \tag{1}$$

where $\beta(N) := \sqrt{\frac{1}{2N} \log \frac{\pi^2 L N^2}{3\delta}}$; $\Delta_0 > 0$ and $\delta > 0$ are the parameters of the attack algorithm. The attacker then gives the post-attack reward $r_t = r_t^0 - \alpha_t$ back to the player. The main idea of the attack algorithm in [9] is to compute the attack value so that the post-attack empirical estimates of the non-target arms are always less than those of the target arm.

However, this attack design only works when the reward space is unbounded and continuous, while in the threat model with binary feedback, the Bernoulli rewards are *discrete* and *bounded*. There are two issues raised by the discrete and bounded requirements. First, the calculated attack value $\alpha_t$ is a real number, which may not make the post-attack reward feedback $r_t$ to be valid (binary). Second, in order to ensure that Eq. (1) is true, the calculated $\alpha_t$ might be larger than 1, which is higher than the maximum attack value in our threat model. In other words, it is impossible to let Eq. (1) hold for all rounds, while such a guarantee was essential for the theoretical analysis in [9]. To overcome these issues, we propose a new attack algorithm against UCB on $L$-armed bandits with binary feedback. It is described in Algorithm 1. The algorithm maintains timestamp $h_a(t)$ for each arm $a$. In round $t$, if the arm pulled by the player, $a_t$, is not the target arm $L$, it first checks the condition $\gamma_t \leq r_t^0$ (Line 6), with $\gamma_t$ computed as

$$\gamma_t = \left[ N_{a_t}(t)\hat{\mu}_{a_t}^0(t) - \sum_{s \in \tau_{a_t}(t-1)} \alpha_s - N_{a_t}(t) \left[ \underline{\mu}_L(t) - \Delta_0 \right]_+ \right]_+, \tag{2}$$

where $[z]_+ = \max(0, z)$ and $\underline{\mu}_L(t) := \hat{\mu}_L(t) - 2\beta(N_L(t))$. In fact, condition $\gamma_t \leq r_t^0$ is equivalent to checking whether there exists a feasible $\alpha_t$ to ensure Eq. (1) holds: if $\gamma_t \leq r_t^0$, with $\alpha_t$ set to be $\lceil \gamma_t \rceil$ (Line 7), Eq. (1) will hold in round $t$. Using a similar analysis as in [9], we can derive an upper bound of $N_{a_t}(t)$ and prove the success of the attack up to round $t$. The algorithm also updates the timestamp $h_{a_t}(t) = t$ for $a_t$ (line 7). If $\gamma_t > r_t^0$, it indicates that there is no feasible $\alpha_t$ that can

ensure Eq. (1). Instead, the algorithm sets $\alpha_t = \lceil \tilde{\gamma}_t \rceil$ with $\tilde{\gamma}_t$ computed as

$$\tilde{\gamma}_t = \left[ N_{a_t}(t)\hat{\mu}_{a_t}^0(t) - \sum_{s \in \tau_{a_t}(t-1)} \alpha_s - N_{a_t}(t) \left[ \underline{\mu}_L(h_{a_t}(t-1)) - \Delta_0 \right]_+ \right]_+, \tag{3}$$

where $h_{a_t}(t-1)$ records the last round that Eq. (1) was satisfied. We can prove such an $\alpha_t$ is always feasible ($\alpha_t \le r_t^0$) and it can ensure that

$$\hat{\mu}_{a_t}(t) \le \hat{\mu}_L(h_{a_t}(t-1)) - 2\beta(N_L(h_{a_t}(t-1))) - \Delta_0. \tag{4}$$

This new inequality always holds for all rounds, which helps guarantee the success of the attack.

**Remark**. Compared with Eq. (1), Eq. (4) uses a more conservative lower bound of $\mu_L$, $\underline{\mu}_L(h_{a_t}(t-1))$, instead of $\underline{\mu}_L(t)$, on the right-hand side of the inequality. We refer to this as *conservative estimation* for the target arm $L$ with respect to arm $a_t$, where the estimated lower bound of $L$ will only be updated when there exists feasible $\alpha_t$ to ensure Eq. (1). Intuitively, it makes the attack algorithm more conservative and less likely to use a small attack value that may result in an invalid large attack value in later rounds. We use an inductive proof to show that there always exists $\alpha_t$ such that Eq. (4) holds while keeping $r_t$ valid (binary). Therefore, conservative estimation addresses the problems posed by binary feedback. This forms the foundation for the attack algorithms tailored for OLTR with click feedback.

### 3.3 Analysis

We first show that our attack algorithm always returns valid binary feedback to the player.

**Lemma 1.** *The post-attack feedback of Algorithm 1 is always valid, i.e., $r_t \in \{0,1\}$ for any t.*

The proof of Lemma 1, which uses an inductive analysis on $\gamma_t, \tilde{\gamma}_t$, can be found in the appendix.

Define $\Delta_a := \mu_a - \mu_L$. We give the following theorem to show the successful attack of Algorithm 1.

**Theorem 1.** *Suppose $T \ge L$ and $\delta \le 1/2$. With probability at least $1 - \delta$, Algorithm 1 misguides the UCB algorithm to choose the target arm $L$ at least $T - (L-1)\left(1 + \frac{3}{\Delta_0^2}\log T\right)$ rounds, using a cumulative attack cost at most*

$$C(T) \le \left(1 + \frac{3}{\Delta_0^2}\log T\right) \sum_{a < L} \left(\Delta_a + \Delta_0 + 4\beta\left(1 + \frac{3}{\Delta_0^2}\log h_a(T)\right)\right).$$

*As $T$ goes to infinity, we have*

$$\lim_{T \to \infty} \frac{C(T)}{\log T} \le O\left(\sum_{a < K} \frac{\Delta_a + \Delta_0}{\Delta_0^2}\right).$$

Compared with Theorem 2 in [9], the $\beta$ term in our cost bound depends on $\log h_a(T)$ instead of $\log T$. Since $\beta$ is a decreasing function and $h_a(T) \le T$, our finite-time cost bound can be larger than that in [9]. However, our asymptotic analysis of the cost suggests that when $T$ is large, such difference becomes negligible. Notice that the attack algorithm in [9] does not have any theoretical guarantee in the binary feedback setting, so this comparison is only meant to show the additional cost potentially caused by the conservative estimation.

## 4 Attacks on Online Learning to Rank

We now move to developing effective attack algorithms for more complicated OLTR settings. Since attacking OLTR also faces the challenges caused by binary click feedback, the attack algorithms in this section rely on our attack design for stochastic bandits with binary feedback in Section 3.

### 4.1 Position-Based Click Model

**Threat Model**. We introduce the threat model for online stochastic ranking with position-based click feedback. In each round $t$, the player chooses a list of $K$ item, $\boldsymbol{a}_t = (a_{1,t}, \cdots, a_{K,t})$ to recommend. The environment generates the pre-attack click feedback $\boldsymbol{r}_t^0 = (r_{1,t}^0, \cdots, r_{K,t}^0) \in \{0,1\}^K$ where $r_{i,t}^0$

---

**Algorithm 2** Attack against the PBM-UCB algorithm

---

1: **Initialization**: Randomly select $K - 1$ items with $L$ to build $\boldsymbol{a}^*$; $h_{l,a}(0) = 1 \ \forall l \in [L] \ \forall a \in \boldsymbol{a}^*$
2: **for** $t = 1, 2, 3, \ldots$ **do**
3:     Observe $\boldsymbol{a}_t, \boldsymbol{r}_t^0$; set $\boldsymbol{\alpha}_t = (\alpha_{1,t}, \cdots, \alpha_{K,t}) = (0, \cdots, 0)$
4:     **for** $i \in [K]$ **do**
5:         **if** $a_{i,t} \notin \boldsymbol{a}^*$ **then**
6:             $\alpha_{i,t} = \texttt{CAL\_ALPHA}(a_{i,t}, r_{i,t}^0, \boldsymbol{a}^*)$
7:         **else**
8:             $\alpha_{i,t} = 0$
9:         **end if**
10:    **end for**
11:    Return $\boldsymbol{r}_t = \boldsymbol{r}_t^0 - \boldsymbol{\alpha}_t$; update $h_{l,a}(t) = h_{l,a}(t-1)$ for all $l \notin \boldsymbol{a}_t, a \in \boldsymbol{a}^*$
12: **end for**

---

**Algorithm 3** `CAL_ALPHA`

---

1: **Input**: item $l$, click feedback $r^0$, item set $\boldsymbol{a}^*$
2: **for** $a \in \boldsymbol{a}^*$ **do**
3:     Calculate $\gamma_t(l, a), \tilde{\gamma}_t(l, a)$ according to Eqs. (6) and (7)
4: **end for**
5: $\gamma_{\max} = \max_{a \in \boldsymbol{a}^*} \gamma_t(l, a), \tilde{\gamma}_{\max} = \max_{a \in \boldsymbol{a}^*} \tilde{\gamma}_t(l, a)$
6: **if** $\gamma_{\max} \leq r^0$ **then**
7:     $\alpha = \lceil \gamma_{\max} \rceil, h_{l,a}(t) = t$ for all $a \in \boldsymbol{a}^*$
8: **else**
9:     $\alpha = \lceil \tilde{\gamma}_{\max} \rceil, h_{l,a}(t) = h_{l,a}(t-1)$ for all $a \in \boldsymbol{a}^*$
10: **end if**
11: **Return** $\alpha$

---

is sampled from a Bernoulli distribution with mean $\kappa_i \mu_{a_{i,t}}$. The attacker then observes $\boldsymbol{a}_t$ and $\boldsymbol{r}_t^0$, and decides the post-attack click feedback $\boldsymbol{r}_t \in \{0, 1\}^K$. The player only receives $\boldsymbol{r}_t$ as the feedback and uses it to decide the next list to recommend, $\boldsymbol{a}_{t+1}$, for round $t + 1$. Without loss of generality, we assume item $L$ is a sub-optimal target item. Similar to Section 3.1, we say the attack is successful after $T$ rounds if the number of target item recommendations is $N_L(T) = T - o(T)$ in expectation or with high probability, while the cumulative attack cost $C(T) = \sum_{t=1}^{T} ||\boldsymbol{r}_t - \boldsymbol{r}_t^0||_1 = o(T)$.

**Attack against PBM-UCB**. We consider the PBM-UCB algorithm in [4] as the online ranking algorithm of the player, which computes the UCB index of each item $a$ as

$$\bar{\mu}_a(t) = \hat{\mu}_a(t-1) + B_a(t) = \hat{\mu}_a(t-1) + \sqrt{\frac{N_a(t-1)(1+\epsilon)\log t}{2\tilde{N}_a(t-1)^2}}, \tag{5}$$

where $\tilde{N}_a(t) := \sum_{s=1}^{t} \sum_{i=1}^{K} \kappa_i I\{a_{i,s} = a\}$ is the position bias-corrected counter, $\hat{\mu}_a(t)$ is the empirical mean of item $a$, and $\epsilon$ is a parameter of the algorithm. The algorithm then chooses the corresponding first $K$ items with the highest UCB indices as the recommendation list.

We propose our attack algorithm against PBM-UCB in Algorithm 2. It works by first randomly taking $K - 1$ items out and making them a set with the target item $L$, denoted as $\boldsymbol{a}^* = \{a_1^*, \cdots, a_{K-1}^*, L\}$. Then, based on the conservative estimation idea from Algorithm 1, it maintains a timestamp $h(l, a)$ for each item $l$ with respect to each $a \in \boldsymbol{a}^*$. The intuition is that, to ensure a similar inequality as Eq. (4) for all rounds, we need to make *conservative estimation* on the lower bounds of $\mu_a$ for all $a \in \boldsymbol{a}^*$. This is handled by Algorithm 3, which maintains the timestamps $h_{l,a}(t)$ for the input item $l$ and outputs the appropriate attack value $\alpha$ on $l$ that can always ensure the required inequality. The value of parameters $\gamma_t(l, a)$ and $\tilde{\gamma}_t(l, a)$ in Algorithm 3 are computed as

$$\gamma_t(l, a) = \left[ N_l(t)\hat{\mu}_l^0(t) - \sum_{s \in \tau_l(t-1)} \alpha_l(s) - N_l(t) \left[ \underline{\mu}_a(t) - \Delta_0 \right]_+ \right]_+, \tag{6}$$

$$\tilde{\gamma}_t(l, a) = \left[ N_l(t)\hat{\mu}_l^0(t) - \sum_{s \in \tau_l(t-1)} \alpha_l(s) - N_l(t) \left[ \underline{\mu}_a(h_{l,a}(t-1)) - \Delta_0 \right]_+ \right]_+. \tag{7}$$

---

**Algorithm 4** Attack against the CascadeUCB algorithm

---

1: **Initialization**: Randomly select $K - 1$ items with $L$ to build $\boldsymbol{a}^*$; $h_{l,a}(0) = 1 \; \forall l \in [L] \; \forall a \in \boldsymbol{a}^*$
2: **for** $t = 1, 2, 3, \ldots$ **do**
3:     Observe $\boldsymbol{a}_t, \boldsymbol{r}_t^0, \mathbf{C}_t$; set $\boldsymbol{\alpha}_t = (\alpha_{1,t}, \cdots, \alpha_{K,t}) = (0, \cdots, 0)$
4:     **if** $\mathbf{C}_t \leq K$ and $a_{\mathbf{C}_t,t} \notin \boldsymbol{a}^*$ **then**
5:         $\alpha_{\mathbf{C}_t,t} = \texttt{CAL\_ALPHA}(a_{\mathbf{C}_t,t}, r_{\mathbf{C}_t,t}^0, \boldsymbol{a}^*)$
6:         **if** $\alpha_{\mathbf{C}_t,t} = 1$ and $\exists i > C_t$ s.t. $a_{i,t} \in \boldsymbol{a}^*$ **then**
7:             $\alpha_{i,t} = -1$
8:         **end if**
9:     **end if**
10:    Return $\boldsymbol{r}_t = \boldsymbol{r}_t^0 - \boldsymbol{\alpha}_t$; update $h_{l,a}(t) = h_{l,a}(t-1)$ for all $l \neq a_{\mathbf{C}_t,t}, a \in \boldsymbol{a}^*$
11: **end for**

---

Notice that Algorithm 2 could also work when there are more than one but less than $K + 1$ target arms (the goal of the attacker becomes misguiding the player to recommend all target arms very often with sublinear cost). The only modification required is to put all of these target arms into $\boldsymbol{a}^*$.

**Analysis**. The following theorem shows the attack against PBM-UCB is successful.

**Theorem 2.** *Suppose $T \geq L$ and $\delta \leq 1/2$. With probability at least $1 - \delta$, Algorithm 2 misguides the PBM-UCB algorithm to recommend the target item $L$ at least $T - (L - K) \left( \frac{1+\epsilon}{2\kappa_K^2 \Delta_0^2} \log T \right)$ rounds, using a cumulative attack cost at most*

$$C(T) \leq \left( \frac{1+\epsilon}{2\kappa_K^2 \Delta_0^2} \log T \right) \sum_{a < L} \left( \Delta_a + \Delta_0 + 4\beta \left( \frac{1+\epsilon}{2\kappa_K^2 \Delta_0^2} \log h_{a,L}(T) \right) \right).$$

*When $T$ goes to infinity, we have*

$$\lim_{T \to \infty} \frac{C(T)}{\log T} \leq O \left( \sum_{a < K} \frac{(1 + \epsilon)(\Delta_a + \Delta_0)}{\kappa_K^2 \Delta_0^2} \right).$$

*Proof sketch.* Whenever $a_{i,t} \notin \boldsymbol{a}^*$ is chosen by the player, there must exist $a \in \boldsymbol{a}^*$ such that $\bar{\mu}_{a_{i,t}}(t) \geq \bar{\mu}_a(t)$. The output $\alpha_{i,t}$ of Algorithm 3 would ensure $\hat{\mu}_{a_{i,t}}(t) \leq \hat{\mu}_a(h_{a_{i,t},a}(t-1)) - 2\beta(N_a(h_{a_{i,t},a}(t-1))) - \Delta_0$, owing to the conservative estimations in $\gamma_t(a_{i,t}, a)$ and $\tilde{\gamma}_t(a_{i,t}, a)$. Combining these two inequalities, we can get $B_{a_{i,t}}(t) - B_a(t) \geq \Delta_0$. With a careful calculation on this inequality and the involved bias-corrected counters, we have $N_{a_{i,t}}(t) \leq \left( \frac{1+\epsilon}{2\kappa_K^2 \Delta_0^2} \log t \right)$. This result holds for any $a_{i,t} \notin \boldsymbol{a}^*$. Thus, we immediately get the bound of $N_L(t)$. The remaining proof for the cost bound will be similar to that of Theorem 1.

Compared with Theorem 1, the asymptotic cost of Algorithm 2 has an additional dependency on $1/\kappa_K^2$, which suggests it may require more cost to achieve a successful attack in the PBM model, though the cost dependency on $T$ is still logarithmic.

### 4.2 Cascade Click Model

**Threat Model**. We introduce the threat model for OLTR with cascade click feedback. In each round $t$, the player chooses a list of $K$ items, $\boldsymbol{a}_t = (a_{1,t}, \cdots, a_{K,t})$ to recommend. The environment generates the pre-attack click feedback $\boldsymbol{r}_t^0 = (r_{1,t}^0, \cdots, r_{K,t}^0) \in \{0, 1\}^K, \|\boldsymbol{r}_t^0\| \leq 1$. Let $\mathbf{C}_t$ denote the position of the clicked item, i.e., $r_{\mathbf{C}_t,t}^0 = 1$ ($\mathbf{C}_t = \infty$ if no item was clicked). The attacker observes $\boldsymbol{a}_t, \boldsymbol{r}_t^0, \mathbf{C}_t$, and decides the post-attack click feedback $\boldsymbol{r}_t \in \{0, 1\}^K, \|\boldsymbol{r}_t\|_1 \leq 1$. The player only receives $\boldsymbol{r}_t$ as the feedback and uses it to decide the next list to recommend, $\boldsymbol{a}_{t+1}$, for round $t + 1$. The goal of the attacker in this setting is the same as that in the PBM model.

**Attack against CascadeUCB**. We propose an attack algorithm against CascadeUCB in Algorithm 4. Similar to the attack against PBM-UCB, it first randomly generates a set of items $\boldsymbol{a}^* = \{a_1^*, \cdots, a_{K-1}^*, L\}$. It also follows the idea of conservative estimation: when the clicked item $a_{\mathbf{C}_t,t}$ does not belong to $\boldsymbol{a}^*$, it calls Algorithm 3 to maintain $h_{a_{\mathbf{C}_t,t},a}$ for all $a \in \boldsymbol{a}^*$ and compute the attack value $\alpha_{\mathbf{C}_t,t}$ based on the conservative estimations. If the output $\alpha_{\mathbf{C}_t,t} = 1$, which means

---

**Algorithm 5** Attack against arbitrary algorithm

---

1: **Initialization**: Randomly select $K - 1$ items with target $L$ to build $\boldsymbol{a}^*$
2: **for** $t = 1, 2, 3, \ldots$ **do**
3:     Observe $\boldsymbol{a}_t, \boldsymbol{r}_t^0$
4:     **for** $i \in [K]$ **do**
5:         $\alpha_{i,t} = 0$
6:         **if** $a_{i,t} \notin \boldsymbol{a}^*$ and $r_{i,t}^0 = 1$ **then**
7:             $p_{i,t} = \max_{a \in \boldsymbol{a}^*} \dfrac{\left[ \hat{\mu}_{a_{i,t}}^0(t) + \beta(N_{a_{i,t}}(t)) - \hat{\mu}_a^0(t) + \beta(N_a(t)) \right]_+}{\hat{\mu}_{a_{i,t}}^0(t) + \beta(N_{a_{i,t}}(t))}$
8:             With prob. $p_{i,t}$, set $\alpha_{i} = 1$
9:         **end if**
10:     **end for**
11:     Return $\boldsymbol{r}_t = \boldsymbol{r}_t^0 - \boldsymbol{\alpha}_t$
12: **end for**

---

the algorithm sets the clicked position to be zero, the player will keep checking the positions after $\mathbf{C}_t$. Since the pre-attack feedback of all items after position $\mathbf{C}_t$ is zero, we need to find the first item $a_{i,t} \in \boldsymbol{a}^*$ after position $\mathbf{C}_t$ and set $\alpha_{i,t} = -1$ ($r_{i,t} = 1$). The empirical means of the items between position $\mathbf{C}_t$ and $i$ that are not in $\boldsymbol{a}^*$ decreases, and the empirical mean of $a_{i,t} \in \boldsymbol{a}^*$ increases. Thus, it does not affect the success of the attack while still making the post-attack feedback $\boldsymbol{r}_t$ valid.

**Analysis**. First, note that there is a mismatch between recommendation and observation in the cascade model: for all $i$ such that $\mathbf{C}_t < i \leq K$, $a_{i,t}$ is recommended but not observed, i.e., there is no new click sample for item $a_{i,t}$ for estimation. We can still follow a similar proof of Theorem 2 to get the upper bound of the number of observations (samples) for $a_{i,t} \notin \boldsymbol{a}^*$, but it is less than the number of recommendations and thus cannot ensure the success of the attack. To tackle this problem, we use a new regret-based analysis on the expected number of recommendations. We can also prove that the post-attack feedback of Algorithm 4 is always valid. Define $p^* := \prod_{i=1}^{K-1} \mu_i$. We give the following theorem of the successful attack against CascadeUCB.

**Theorem 3.** *Suppose $T \geq L$ and $\delta \leq 1/2$. With probability at least $1 - \delta$, Algorithm 2 misguides the CascadeUCB algorithm to choose the target arm at least $T - (L - K) \left( \frac{12}{p^* \Delta_0^2} \log T \right)$ rounds in expectation. Its cumulative attack cost is at most*

$$C(T) \leq \left( 1 + \tfrac{3}{\Delta_0^2} \log T \right) \sum_{a < K} \left( \Delta_a + \Delta_0 + 4\beta \left( 1 + \tfrac{3}{\Delta_0^2} \log h_{a,L}(T) \right) \right).$$

*As $T$ goes to infinity, we have*

$$\lim_{T \to \infty} \frac{C(T)}{\log T} \leq O \left( \sum_{a < K} \frac{\Delta_a + \Delta_0}{\Delta_0^2} \right).$$

*Proof sketch.* As mentioned above, we use a new regret-based analysis. The intuition is that the regret caused by any suboptimal item is the product of its reward gap with respect to the optimal list and its expected number of recommendations. If we know the regret upper bound and the reward gap lower bound, we can derive the upper bound of the expected number of recommendations. To do so, we first show that the post-attack problem can be viewed as a problem with a known reward gap lower bound. This can be obtained by $\hat{\mu}_{a_{i,t}}(t) \leq \hat{\mu}_L(h_{L_{i,t},a}(t-1)) - 2\beta(N_a(h_{a_{i,t},L}(t-1))) - \Delta_0$ for any $a_{i,t} \notin \boldsymbol{a}^*$, which indicates the post-attack expected reward gap between $a_{i,t}$ and $L$ is always larger than $\Delta_0$. Then, the lower bound of the reward gap of $a_{i,t}$ with respect to the optimal list is $p^* \Delta_0$. Based on the regret upper bound $12 \log T / \Delta_0$ given in [5], we can get the upper bound of the expected number of recommendations $\frac{12}{p^* \Delta_0^2} \log T$. For the remaining cost analysis, since the attack cost only depends on the observed items, we can still follow the proof of Theorem 2.

## 4.3 General Attacks on OLTR with General Click Model

We have provided attack strategies against UCB-based OLTR algorithms under two specific click models. A natural follow-up question is whether there exists an attack strategy that can attack any

OLTR algorithm under the general click model. To answer this question in what follows, we design an attack strategy that can misguide any OLTR algorithm without knowing the underlying algorithm. However, it may pay more cost (still sublinear) than the others since it cannot take advantage of the details of the algorithm to make fine-tuned adjustments.

**Threat Model**. We consider the threat model for OLTR with general click feedback. In each round $t$, the player chooses a list of $K$ items, $\boldsymbol{a}_t = (a_{1,t}, \cdots, a_{K,t})$ to recommend. The environment generates the pre-attack click feedback $\boldsymbol{r}_t^0 = (r_{1,t}^0, \cdots, r_{K,t}^0) \in \{0,1\}^K$, $\boldsymbol{r}_t^0 \in \mathcal{R}_c$, where $\mathcal{R}_c$ is the feasible feedback space of click model $c$. The attacker observes $\boldsymbol{a}_t, \boldsymbol{r}_t^0$, and decides the post-attack click feedback $\boldsymbol{r}_t \in \{0,1\}^K$, $\boldsymbol{r}_t \in \mathcal{R}_c$. The attack should be aware of $\mathcal{R}_c$; otherwise, ensuring valid post-attack feedback is impossible. The player only receives $\boldsymbol{r}_t$ as the feedback and uses it to decide the next list to recommend, $\boldsymbol{a}_{t+1}$. The goal of the attacker is the same as that in the PBM model.

**General Attack against Arbitrary OLTR Algorithm**. We propose an attack algorithm against arbitrary OLTR algorithms in Algorithm 5. Similar to the attack against PBM-UCB, it first randomly generates a set of items $\boldsymbol{a}^* = \{a_1^*, \cdots, a_{K-1}^*, L\}$. In each round, for each clicked item $a_{i,t} \notin \boldsymbol{a}^*$, the algorithm calculates an attack probability $p_{i,t}$ and uses that to decide whether its feedback needs to be changed to unclicked ($r_{i,t} = 0$). We prove that $p_{i,t}$ is actually an estimated upper bound of $\Delta_i/\mu_i$, thus by such probabilistic feedback perturbations, the algorithm makes all items outside $\boldsymbol{a}^*$ be worse than the items inside $\boldsymbol{a}^*$, which guarantees the success of the attack. We consider a general assumption for the OLTR algorithm to be attacked.

**Assumption 1.** *The OLTR algorithm chooses suboptimal items no more than $R(T) = o(T)$ times for $T$ rounds,*

Notice that both PBM-UCB and CascadeUCB satisfy the assumption with $R(T) = O(\log T)$. We give the following theorem to show the success of our general attack algorithm.

**Theorem 4.** *Suppose $T \geq L$ and $\delta \leq 1/2$. With probability at least $1 - \delta$, Algorithm 5 misguides arbitrary OLTR algorithm that satisfies Assumption 1 to choose the target item at least $T - R(T)$ rounds in expectation. Its cumulative attack cost is at most*

$$C(T) \leq O\left(\sum_{a<L}(\Delta_a + 4\beta(1))R(T)\right).$$

Compared with the results in the PBM and cascade models, the $\beta()$ term in the cumulative cost of Algorithm 5 is $\beta(1)$, which can be much larger than the others. Also, the asymptotic costs of Algorithm 2 and Algorithm 4 is independent of $\beta$, while the asymptotic cost of Algorithm 5 depends on $\beta$, showing that Algorithm 5 is more costly than attack strategies specific to OLTR algorithms.

## 5 Experiments

We conduct experiments using both synthetic and real data (MovieLens 20M dataset [26]). Due to space limitations, we report only the results of OLTR using the position-based model. We use $\epsilon = 0.1$ for the PBM-UCB algorithm. For the synthetic data, we take $L = 16, K = 8, T = 100,000$; $\{\mu_i\}_{i=1}^L$ are sampled from uniform distribution $U(0,1)$ for Figure 1a, and from $U(0,x)$ for Figure 1b. For the real data, we take $L = 100, K = 10, T = 100,000$; $\{\mu_i\}_{i=1}^L$ are extracted according to [2].

Using synthetic data, we first study how the algorithm's performance is influenced by the algorithmic and problem parameters. As shown in Figure 1a, the cost decreases with an increase in $\Delta_0$, aligning with our observations in Theorem 2. Figure 1b shows that the cost increases as $x$ increases, which suggests that our algorithm pays more costs when $\Delta_a$ is large. We then compare our algorithm with two baselines: $\texttt{Trivial}_{\text{all}}$ attacks all arms except the target arm as long as the attack is valid; $\texttt{Trivial}_{\text{set}}$ first randomly takes $K - 1$ arms out to generate a set $\boldsymbol{a}^*$ and then attacks all arms outside the set as long as the attack is valid. Figure 2 shows that $\texttt{Trivial}_{\text{all}}$ algorithm cannot successfully attack PBM-UCB even with linear costs. Our algorithm and $\texttt{Trivial}_{\text{set}}$ have similar performance on the chosen ratio of the target arm as shown in Figures 2a and 2c, which is the chosen time of the target arm divided by the current round. However, our algorithm pays $50\%$ and $40\%$ less cost than $\texttt{Trivial}_{\text{set}}$ in Figures 2b and 2d, respectively, which validates the necessity of our attack design.

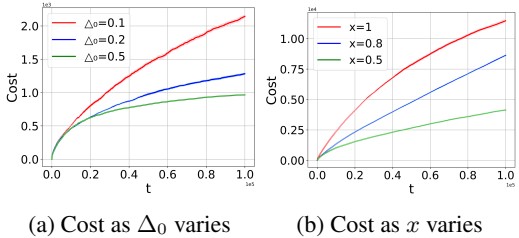

(a) Cost as $\Delta_0$ varies     (b) Cost as $x$ varies

Figure 1: Impact of algorithmic and problem parameters on PBM-UCB.

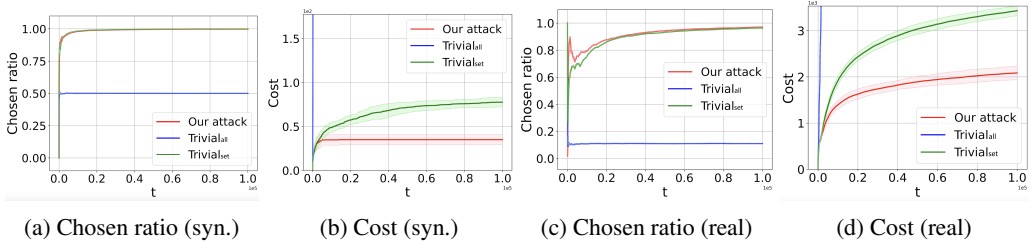

(a) Chosen ratio (syn.)     (b) Cost (syn.)     (c) Chosen ratio (real)     (d) Cost (real)

Figure 2: Companion with baseline algorithms.

## 6   Concluding Remarks

This paper presents the first study on adversarial attacks against OLTR with click feedback. We propose attack strategies that can successfully mislead various OLTR algorithms across multiple click models. One limitation of our work is that it cannot be applied to feedback models without clicks, such as the top-$K$ feedback [13]. Although we provide theoretical results for all these strategies, as discussed in Section 3.3, the finite-time cost results might be further improved by a more fine-grained analysis on $h_a(t)$. Additionally, there is no known lower bound for the cumulative attack cost, even for the stochastic bandit setting in the literature, making it unclear whether our attack strategies are (asymptotically) order-optimal. This study opens up several future directions. One is to design attack strategies for other OLTR algorithms and feedback models. Lastly, our study on the vulnerability of existing OLTR algorithms inspires the design of robust OLTR algorithms against adversarial attacks.

## Acknowledgments and Disclosure of Funding

Shuai Li is supported by National Key Research and Development Program of China (2022ZD0114804) and National Natural Science Foundation of China (62376154, 62006151, 62076161). M. Hajiesmaili's work is supported by CAREER-2045641, CNS-2102963, CNS-2106299, and CPS-2136199. Wierman is supported by NSF grants CNS-2146814, CPS-2136197, CNS-2106403, NGSDI-2105648.

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

# Appendix

## A   Proofs

### A.1   Proof of Lemma 1

*Proof.* When $\gamma_t \leq r_t^0$, it is easy to check $\alpha_t = \lceil \gamma_t \rceil \leq r_t^0$, thus $r_t \in \{0, 1\}$.

When $\gamma_t > r_t^0$, if $\tilde{\gamma}_t \leq r_t^0$, we will have $\alpha_t = \lceil \tilde{\gamma}_t \rceil \leq r_t^0$ and $r_t \in \{0, 1\}$. Since $\tilde{\gamma}_t$ depends on $a_t$, our goal is to show $\tilde{\gamma}_t \leq r_t^0$ for any $a_t$. Consider any arm $a \neq K$. We denote $t_{a,j}$ as the $j$-th time that the UCB algorithm played $a$. Since the UCB algorithm would play each arm one time in the beginning, we have

$$\tilde{\gamma}_{t_{a,1}} \leq \left[ N_a(t_{a,1}) \hat{\mu}_a^0(t_{a,1}) \right]_+ = \left[ r_{t_{a,1}}^0 \right]_+ \leq r_{t_{a,1}}^0. \tag{8}$$

Next, we want to show that if $\tilde{\gamma}_{t_{a,j}} \leq r_{t_{a,j}}^0$, $\tilde{\gamma}_{t_{a,j+1}} \leq r_{t_{a,j+1}}^0$. Since $\tilde{\gamma}_t$ also depends on $h_{a_t}(t-1)$, we consider two cases:

1) $\gamma_{t_{a,j}} \leq r_{t_{a,j}}^0$. In this case, we have $h_a(t_{a,j+1} - 1) = t_{a,j}$. We can bound $\tilde{\gamma}_{t_{a,j+1}}$ using $\gamma_{t_{a,j}}$:

$$\tilde{\gamma}_{t_{a,j+1}} \leq \gamma_{t_{a,j}} + r_{t_{a,j+1}}^0 - \alpha_{t_{a,j}} - \left[ \underline{\mu}_L(t_{a,j}) - \Delta_0 \right]_+ \leq r_{t_{a,j+1}}^0, \tag{9}$$

where the second inequality is due to $\gamma_{t_{a,j}} \leq \alpha_{t_{a,j}}$.

2) $\gamma_{t_{a,j}} > r_{t_{a,j}}^0$. In this case, we have $h_a(t_{a,j+1} - 1) = h_a(t_{a,j} - 1)$. We can bound $\tilde{\gamma}_{t_{a,j+1}}$ using $\tilde{\gamma}_{t_{a,j}}$

$$\tilde{\gamma}_{t_{a,j+1}} \leq \tilde{\gamma}_{t_{a,j}} + r_{t_{a,j+1}}^0 - \alpha_{t_{a,j}} - \left[ \underline{\mu}_L(h_a(t_{a,j} - 1)) - \Delta_0 \right]_+ \leq r_{t_{a,j+1}}^0, \tag{10}$$

where the second inequality is due to $\tilde{\gamma}_{t_{a,j}} \leq r_{t_{a,j}}^0$ and $\tilde{\gamma}_{t_{a,j}} \leq \alpha_{t_{a,j}}$.

Since $\tilde{\gamma}_{t_{a,1}} \leq r_{t_{a,1}}^0$, by induction, we have $\tilde{\gamma}_{t_{a,j}} \leq r_{t_{a,j}}^0$ for any $a, j$, which concludes the proof. □

### A.2   Proof of Theorem 1

*Proof.* We use the following two lemmas to prove Theorem 1. The proof is similar to those of Lemma 5 and Lemma 6 in [9], while as discussed in Section 3, the algorithm in [9] relies on Eq. (1) that may not hold due to the binary feedback, and our proof is mainly based on Eq. (4) that can be ensured by the conservative estimations. Define event $E := \{ \forall i, \forall t > L : |\hat{\mu}_i^0(t) - \mu_i| < \beta(N_i(t)) \}$. With Hoeffding's inequality, it is easy to prove for $\delta \in (0, 1)$, $\mathbb{P}(E) > 1 - \delta$.

**Lemma 2.** *Assume event $E$ holds and $\delta \leq 1/2$. For any $a \neq L$ and any $t \geq L$, we have*

$$N_a(t) \leq \min\{N_L(t), 1 + \frac{3}{\Delta_0^2} \log t\} \tag{11}$$

*Proof.* Fix some $t \geq L$. We consider $a_t = a \neq L$ and denote $t' = \max\{s < t : a_s = a\}$ as the previous round that $a$ was pulled. Since arm $a$ should be pulled at least once by the UCB algorithm, we have $t' \geq 1$, and the attack in round $t'$ would ensure

$$\hat{\mu}_a(t') \leq \underline{\mu}_L(h_a(t')) - \Delta_0. \tag{12}$$

Since arm $a$ was pulled in round $t$, we know its UCB value must be greater than that of target arm $L$:

$$\hat{\mu}_a(t-1) + \frac{3}{2}\sqrt{\frac{\log(t)}{N_a(t-1)}} \geq \hat{\mu}_L(t-1) + \frac{3}{2}\sqrt{\frac{\log(t)}{N_L(t-1)}}. \tag{13}$$

Considering that $t'$ and $t$ are two consecutive rounds when arm $a$ was pulled, we have $\hat{\mu}_a(t-1) = \hat{\mu}_a(t')$ and $N_a(t-1) = N_a(t')$. Rearranging Eq. (13), we get

$$\frac{3}{2}\sqrt{\frac{\log(t)}{N_a(t')}} - \frac{3}{2}\sqrt{\frac{\log(t)}{N_L(t-1)}} \geq \hat{\mu}_L(t-1) - \hat{\mu}_a(t') \geq \hat{\mu}_L(t-1) - (\underline{\mu}_L(h_a(t')) - \Delta_0) \geq \Delta_0, \tag{14}$$

where the second inequality comes from Equation (12). Since $\Delta_0 > 0$, we have

$$N_a(t) = N_a(t') + 1 \leq N_L(t-1) = N_L(t). \tag{15}$$

Also, since $\frac{3}{2}\sqrt{\frac{\log(t)}{N_L(t-1)}} > 0$, we have $\frac{3}{2}\sqrt{\frac{\log(t)}{N_a(t')}} > \Delta_0$, which implies

$$N_a(t) = N_a(t') + 1 \leq 1 + \frac{3}{\Delta_0^2}\log(t). \tag{16}$$

$\square$

**Lemma 3.** *Assume event $E$ holds and $\delta \leq 1/2$.*

*1) For any $t \geq L$, the cumulative attack cost to any fixed arm $a \neq L$ can be bounded as:*

$$\sum_{s \in \tau_a(t)} \alpha_s \leq N_a(t)\left(\Delta_a + \Delta_0 + 3\beta(N_L(h_a(t)) + \beta(N_a(t)))\right) + 1 \tag{17}$$

*2) When $t$ goes to infinity, we have*

$$\lim_{t \to \infty} \frac{\sum_{s \in \tau_a(t)} \alpha_s}{\log t} \leq \frac{3}{\Delta_0^2}(\Delta_a + \Delta_0) \tag{18}$$

*Proof.* Fix any arm $a \neq L$. By the definitions of $\gamma_t$ and $\tilde{\gamma}_t$, it follows that:

$$\sum_{s \in \tau_a(t)} \alpha_s \leq \sum_{s \in \tau_a(t-1)} \alpha_s + 1 \tag{19}$$

$$\leq N_a(t)\hat{\mu}_a^0(t) - N_a(t)\left[\underline{\mu}_L(h_a(t-1)) - \Delta_0\right] + 1 \tag{20}$$

$$\leq N_a(t)\left[\Delta_a + 3\beta(N_L(h_a(t))) + \beta(N_a(t))\right] + 1, \tag{21}$$

where the last inequality is due to the decrease of $\beta$.

For the asymptotic result, combining Lemma 2 with Equation (17), we have

$$\lim_{t \to \infty} \frac{\sum_{s \in \tau_a(t)} \alpha_s}{\log t} \leq \lim_{t \to \infty} \frac{N_a(t)}{\log(t)}\left[\Delta_a + 3\beta(N_L(h_a(t))) + \beta(N_a(t))\right] + \frac{1}{\log(t)} \tag{22}$$

$$\leq \lim_{t \to \infty}\left(\frac{1}{\log(t)} + \frac{3}{\Delta_0^2}\right)\left[\Delta_a + 3\beta(N_L(h_a(t))) + \beta(N_a(t))\right] + \frac{1}{\log(t)}. \tag{23}$$

It is easy to check that $\lim_{t \to \infty} \beta(t) = 0$. Hence, to get the asymptotic cost bound in the lemma, we need to prove $\lim_{t \to \infty} \beta(N_L(h_a(t))) = 0$. We find $\lim_{t \to \infty} h_a(t) = t$ is a sufficient condition for it; in other words, $h_a(t)$ should be always updated when $t$ goes to infinity. To obtain this, we consider two cases of $\gamma_t$.

1) If $\underline{\mu}_L(t) - \Delta_0 \geq 0$, we have

$$\lim_{t \to \infty} \gamma_t = \lim_{t \to \infty}\left(N_a(t)\hat{\mu}_a^0(t) - \sum_{s \in \tau_a(t-1)} \alpha_s - N_a(t)(\hat{\mu}_L(t) - 2\beta(N_L(t)) - \Delta_0)\right)$$

$$= \lim_{t \to \infty}\left(N_a(t)\hat{\mu}_a^0(t) - N_a(t')(\hat{\mu}_L(h_a(t')) - 2\beta(N_L(h_a(t'))) - \Delta_0) + r_t^0 - \alpha_{t'}\right.$$

$$\left. + N_a(t')(\hat{\mu}_L(h_a(t')) - 2\beta(N_L(h_a(t'))) - \Delta_0) - N_a(t)(\hat{\mu}_L(t) - 2\beta(N_L(t)) - \Delta_0)\right)$$

$$\leq \lim_{t \to \infty}\left(r_t^0 - \alpha_{t'} + N_a(t')(\hat{\mu}_L(h_a(t')) - 2\beta(N_L(h_a(t'))) - \Delta_0) - N_a(t)(\hat{\mu}_L(t) - 2\beta(N_L(t)) - \Delta_0)\right)$$

$$\leq r_t^0 + \lim_{t \to \infty}\left(N_a(t')(\hat{\mu}_L(h_a(t')) - \hat{\mu}_L(t) - 2\beta(N_L(h_a(t')))) - (\hat{\mu}_L(t) - \Delta_0)\right)$$

$$\leq r_t^0.$$

The first inequality is due to the attack at round $t'$. The last inequality is due to the confidence radius based on $\beta(N_L(h_a(t')))$ and $\beta$ is decreasing.

2) If $\underline{\mu}_L(t)-\Delta_0 < 0$, we have $\hat{\mu}_L(t) < 2\beta(N_L(t))+\Delta_0$ and $\hat{\mu}_L(h_a(t')) \leq \hat{\mu}_L(t)+2\beta(N_L(h_a(t')))$.

$$
\begin{aligned}
\lim_{t\to\infty} \gamma_t &= r_t^0 + \lim_{t\to\infty} (N_a(t')\hat{\mu}_a^0(t') - \sum_{s\in\tau_a(t-1)} \alpha_s) \\
&= r_t^0 + \lim_{t\to\infty} N_a(t')\hat{\mu}_a(t') \\
&\leq r_t^0 + \lim_{t\to\infty} N_a(t')(\hat{\mu}_L(h_a(t')) - 2\beta(N_L(h_a(t'))) - \Delta_0) \\
&\leq r_t^0 + \lim_{t\to\infty} N_a(t')2\beta(N_L(t)) \\
&= r_t^0.
\end{aligned}
$$

Since in both cases, $\gamma_t \leq r_t^0$, we have $h_a(t) = t$ when $t$ goes to infinity. $\qquad\square$

With Lemma 2 and Lemma 3, the proof is completed by summing the corresponding upper bounds over all non-target arms $a < L$, $\qquad\square$

### A.3 Proof of Theorem 2

*Proof.* As for Algorithm 1, we first need to prove the post-attack feedback of Algorithm 2 is always valid. It is equivalent to showing that the output $\alpha$ of Algorithm 3 is a valid attack value on the input pre-attack feedback $r^0$, i.e., $\alpha \leq r^0$. Similar to the proof of Lemma 1, we consider two cases: when $\gamma_{\max} \leq r^0$, $\alpha = \lceil\gamma_{\max}\rceil \leq r^0$; when $\gamma_{\max} < r^0$, we can use the same inductive proof of Lemma 1 to show that $\alpha = \lceil\tilde{\gamma}_{\max}\rceil \leq r^0$. Thus, the post-attack feedback $r^0 - \alpha$ is always valid.

Fix some $t \geq L$ such that $a_{i,t} \notin \boldsymbol{a}^*$. We denote $t' = \max\{s < t : a_{i,t} \in \boldsymbol{a}_s\}$ as the previous round that $a_{i,t}$ was chosen. With the conservative estimations in $\gamma_t(a_{i,t}, a)$ and $\tilde{\gamma}_t(a_{i,t}, a)$, the output $\alpha_{i,t}$ of Algorithm 3 could ensure

$$
\hat{\mu}_{a_{i,t}}(t) \leq \hat{\mu}_a(h_{a_{i,t},a}(t-1)) - 2\beta(N_a(h_{a_{i,t},a}(t-1))) - \Delta_0, \tag{24}
$$

for any $a \in \boldsymbol{a}^*$. Since $a_{i,t}$ is chosen by the PBM-UCB algorithm, there must exist $a \in \boldsymbol{a}^*$ with its UCB value less than that of $a_{i,t}$:

$$
\hat{\mu}_a(t-1) + B_a(t) \leq \hat{\mu}_{a_{i,t}}(t-1) + B_{a_{i,t}}(t). \tag{25}
$$

Since $t'$ and $t$ are two consecutive rounds when item $a_{i,t}$ was chosen, we have $\hat{\mu}_{a_{i,t}}(t-1) = \hat{\mu}_{a_{i,t}}(t'), N_{a_{i,t}}(t-1) = N_{a_{i,t}}(t')$. Rearranging Eq. (25), we have

$$
B_{a_{i,t}}(t) - B_a(t) \geq \hat{\mu}_a(t-1) - \hat{\mu}_{a_{i,t}}(t') \geq \Delta_0, \tag{26}
$$

where the last inequality is due to Eq. (24). Since $B_{a_{i,t}}(t) \geq \Delta_0$, with the definition of $B_{a_{i,t}}(t)$ in Eq. (5), we have

$$
N_{a_{i,t}}(t) \leq \frac{1+\epsilon}{2\kappa_K^2\Delta_0^2} \log t, \tag{27}
$$

for any $a_{i,t} \notin \boldsymbol{a}^*$. Thus, for the target item $L$,

$$
N_L(T) \geq T - \sum_{l\notin\boldsymbol{a}^*} N_l(T) \geq T - (L-K)\left(\frac{1+\epsilon}{2\kappa_K^2\Delta_0^2} \log T\right), \tag{28}
$$

which guarantees the chosen time of the target item.

For the cumulative attack cost analysis, we consider any arm $a_{i,t} \notin \boldsymbol{a}^*$. Since Eq. (24) holds for any $a \in \boldsymbol{a}^*$, we take $a = L$ and get

$$
\sum_{s\in\tau_{a_{i,t}}(t)} \alpha_{a_{i,t},s} \leq N_{a_{i,t}}(t)\left[\Delta_{a_{i,t}} + \Delta_0 + 3\beta(N_L(h_{a_{i,t},L}(t))) + \beta(N_{a_{i,t}}(t))\right] \tag{29}
$$

$$
\leq N_{a_{i,t}}(t)\left[\Delta_{a_{i,t}} + \Delta_0 + 4\beta(N_{a_{i,t}}(h_{a_{i,t},L}(t)))\right], \tag{30}
$$

where the second line is due to $N_{a_{i,t}}(t) \leq N_L(t), h_{a_{i,t},L}(t) \leq N_{a_{i,t}}(t)$, and $\beta$ is decreasing. Based on this inequality, we can follow the same steps in the proof of Lemma 3 to derive the upper bound of the cumulative attack cost for Algorithm 2. $\qquad\square$

## A.4 Proof of Theorem 3

*Proof.* Since Algorithm 4 also calls Algorithm 3, we can use the same proof in Appendix A.3 to show its post-attack feedback is always valid. Also, the output $\alpha_{i,t}$ of Algorithm 3 would still ensure Eq. (24) for any $a_{i,t} \notin \boldsymbol{a}^*, a \in \boldsymbol{a}^*$. Based on this, we can consider the post-attack problem as a cascading bandit problem with known expected reward gaps, where the expected reward of any item $l \notin \boldsymbol{a}^*$ is less than that of the target item $L$ by at least $\Delta_0$. Since $L$ is the worst item in $\boldsymbol{a}^*$, with the regret analysis of CascadeUCB in [5], we know the regret caused by any list containing $l \notin \boldsymbol{a}^*$ is bounded by

$$R_l(T) \leq \frac{12}{\Delta_0} \log T. \tag{31}$$

We also know the lower bound of the one-round instantaneous regret caused by any list containing $l$ is $p^* \Delta_0$, where $p^* := \prod_{i=1}^{K-1} \mu_i$. We can then write the lower bound of $R_l(T)$ as

$$R_l(T) \geq p^* \Delta_0 \cdot \mathbb{E}[N_l(T)]. \tag{32}$$

Combining Eq. (31) and Eq. (32), we obtain the upper bound of the expected number of recommendations of item $l$:

$$\mathbb{E}[N_l(T)] \leq \frac{12}{p^* \Delta_0^2} \log T. \tag{33}$$

Thus, for the target item $L$,

$$\mathbb{E}[N_L(T)] \geq T - \sum_{l \notin \boldsymbol{a}^*} \mathbb{E}[N_l(T)] \geq T - (L - K) \left( \frac{12}{p^* \Delta_0^2} \log T \right). \tag{34}$$

For the remaining cost analysis, since the attack cost only depends on the observed items, we can still follow the proof of Theorem 2: we need to change the confidence radius $B_a(t)$ in Eq. (5) to the typical form $\frac{3}{2} \sqrt{\frac{\log t}{N_a(t-1)}}$, then derive the corresponding cost upper bound. $\qquad\square$

## A.5 Proof of Theorem 4

*Proof.* Since Algorithm 5 only attacks when $r_{i,t}^0 = 1$, it is easy to verify its post-attack feedback is valid. The key step for proving its successful attack is to show that $p_{i,t}$ is an upper bound of $\Delta_{a_{i,t}}/\mu_{a_{i,t}}$: if it is true, by the probabilistic attack with probability $p_{i,t}$, Algorithm 5 makes a post-attack bandit problem where the expected reward of any item $l \notin \boldsymbol{a}^*$ is worse than that of target item $L$, which guarantees the number of target arm pulls is $T - R(T)$. Actually, we have

$$\frac{\Delta_{a_{i,t}}}{\mu_{a_{i,t}}} = \frac{\mu_{a_{i,t}} - \mu_L}{\mu_{a_{i,t}}} = 1 - \frac{\mu_L}{\mu_{a_{i,t}}}, \tag{35}$$

which increases as $\mu_{a_{i,t}}$ increases. Replacing $\mu_{a_{i,t}}$ with $\hat{\mu}_{a_{i,t}}^0(t) + \beta(N_{a_{i,t}}(t))$,

$$\frac{\Delta_{a_{i,t}}}{\mu_{a_{i,t}}} \leq \frac{\hat{\mu}_{a_{i,t}}^0(t) + \beta(N_{a_{i,t}}(t)) - \mu_L}{\hat{\mu}_{a_{i,t}}^0(t) + \beta(N_{a_{i,t}}(t))} \leq \frac{\left[ \hat{\mu}_{a_{i,t}}^0(t) + \beta(N_{a_{i,t}}(t)) - \hat{\mu}_L^0(t) + \beta(N_L(t)) \right]_+}{\hat{\mu}_{a_{i,t}}^0(t) + \beta(N_{a_{i,t}}(t))}, \tag{36}$$

where $\hat{\mu}_{a_{i,t}}^0(t) + \beta(N_{a_{i,t}}(t)) \geq \mu_{a_{i,t}}$ and $\hat{\mu}_L^0(t) + \beta(N_L(t)) \geq \mu_L$ under event $E$. Since the last term is the definition of $p_{i,t}$, we have proved it is a high-probability upper bound of $\Delta_{a_{i,t}}/\mu_{a_{i,t}}$. For the cost analysis, we have

$$C(T) \leq \sum_{i,t} \left[ \hat{\mu}_{a_{i,t}}^0(t) + \beta(N_{a_{i,t}}(t)) - \hat{\mu}_L^0(t) + \beta(N_L(t)) \right]_+ \mathbb{I}\{a_{i,t} \notin \boldsymbol{a}^*\} \tag{37}$$

$$\leq \sum_{i,t} \left[ \mu_{a_{i,t}} + 2\beta(N_{a_{i,t}}(t)) - \mu_L + 2\beta(N_L(t)) \right]_+ \mathbb{I}\{a_{i,t} \notin \boldsymbol{a}^*\} \tag{38}$$

$$\leq O \left( \sum_{a < L} (\Delta_a + 4\beta(1)) R(T) \right), \tag{39}$$

which concludes the proof. $\qquad\square$

# B   Additional Experiments

## B.1   Synthetic Data

We first consider adversarial attacks on 2-armed stochastic bandits with binary feedback. The rewards of arms 1 and 2 follow Bernoulli distributions with means $\mu_1$ and $\mu_2$ ($\mu_1 > \mu_2$). We designate arm 2 as the target arm. We take $T = 10,000, \delta = 0.1, \Delta_0 = 0.1$. We set $\mu_1 = 0.85$ and sample the value of $\mu_2$ from a uniform distribution $U(0.05, 0.15)$. We compare Algorithm 1 with a modified algorithm from [9], which attacks every non-target arm whenever the calculated $\alpha_t > 0$ and the pre-attack feedback is one. Note that this modified algorithm lacks a theoretical guarantee. We show the costs and target arm chosen times relative to a trivial baseline, which sets the post-attack feedback of all non-target arms to zero as long as the pre-attack feedback is one. Figure 3 demonstrates that our algorithm is more efficient than the modified one, incurring fewer costs for a greater number of target arm selections.

We also conduct experiments under the cascade model with synthetic data. We take $L = 16, K = 8, T = 100,000$, and $\{\mu_i\}_{i=1}^{L}$ are sampled from a uniform distribution $U(0, 1)$. We compare Algorithm 4 with the same baselines introduced in Section 5. Figure 4 shows that $\texttt{Trivial}_{\text{all}}$ algorithm suffers an extremely high cumulative cost, while it cannot misguide the agent to recommend the target arm very often. On the other hand, our algorithm and $\texttt{Trivial}_{\text{set}}$ algorithm can successfully attack CascadeUCB and perform similarly on the chosen ratio of the target arm. However, our algorithm pays about $30\%$ less cost than $\texttt{Trivial}_{\text{set}}$.

## B.2   Real Data

As discussed in Section 5, we have shown results under the position-based model with real data. We now show the experimental results with real data (MovieLens 20M dataset) under the cascade model. We take $L = 100, K = 10, T = 100,000$; $\{\mu_i\}_{i=1}^{L}$ are extracted according to [2]. Similarly, we compare the chosen ratio and cumulative cost of our algorithm with $\texttt{Trivial}_{\text{all}}$ and $\texttt{Trivial}_{\text{set}}$. Figure 5 shows that $\texttt{Trivial}_{\text{all}}$ cannot successfully attack CascadeUCB as the chosen ratio of the target arm is very low and it suffers a linear cost. Algorithm 4 and $\texttt{Trivial}_{\text{set}}$ have similar performance on the chosen ratio of the target arm. However, our algorithm dramatically decreases the cost by more than $50\%$, which indicates that it is more effective than $\texttt{Trivial}_{\text{set}}$.

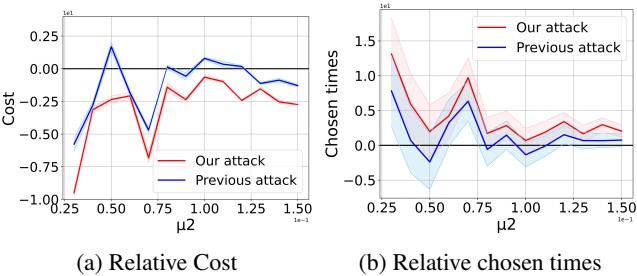

(a) Relative Cost                    (b) Relative chosen times

Figure 3: Attacks against UCB on 2-armed bandits with binary feedback.

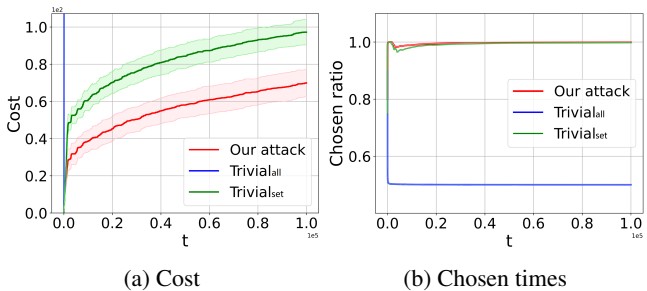

(a) Cost                    (b) Chosen times

Figure 4: Attacks against CascadeUCB with synthetic data.

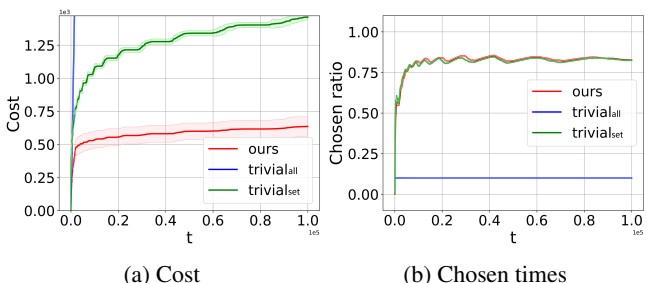

(a) Cost                    (b) Chosen times

Figure 5: Attacks against CascadeUCB with real data.

