# OpenReview forum: "Adversarial Attacks on Online Learning to Rank with Click Feedback"
_NeurIPS.cc/2023/Conference — NeurIPS 2023 poster_

### Official Review · Reviewer_E7da · 2023-07-05

**Soundness:** 4 excellent
**Presentation:** 4 excellent
**Contribution:** 4 excellent
**Rating:** 6
**Confidence:** 5

**Summary:**

This paper presented a study of reward poisoning attacks on UCB bandit algorithms with binary feedback and applies the designed attack to the online-learning-to-rank (OLTR) problem. The binary reward structure poses unique challenges to reward poisoning attacks, since the attacker needs to ensure the perturbed reward is still binary. The authors proposed an attack algorithm that achieves that. Based on that, the paper studied reward poisoning attacks in OLTR. The attacker is limited to change the click feedback, which is required to be binary before and after attack. Two threat models are considered - in the position-based click feedback scenario, the authors proposed attack strategy targeting the PBM-UCB algorithm; and in the cascade-click model, attack strategy against CascadeUCB is proposed. Both achieve T-O(log T) target item selections with only O(log T) attack cost. The paper also designed a general attack strategy for arbitrary click models that achieve O(log T) sub-optimal item selections. Finally, the paper performed empirical evaluations of the attacks.

**Strengths:**

(1). The paper first studied data poisoning attacks on ranking algorithms, which is timely topic following prior works that study reward poisoning attacks on multi-armed bandit. Although the idea behind the design of the attack algorithm is similar in nature, poisoning ranking algorithms poses unique challenges compared to bandits - the attack needs to ensure the poisoned reward is still binary.

(2). The theoretical analysis of this paper is solid. First, an efficient attack targeting the UCB bandit algorithm is proposed, which guarantees that the poisoned reward is still binary. Then two theoretical results are presented to show that UCB-style ranking algorithms are susceptible to poisoning attacks in both position-click and cascade-click scenarios. Finally, the paper also provided a general attack for general click model ranking algorithms.

(3). The paper performed extensive empirical evaluations of the proposed attack strategies. The results are convincing.

**Weaknesses:**

(1). The assumption made in section 4.3 is hidden in Theorem 4 - the ranking algorithm needs to guarantee O(log T) sub-optimal item selections. This assumption is critical and should not be hidden inside the theorem. Instead, the authors should consider carefully defining the assumption.

(2). The Theorem 4 is not intriguing enough. In particular, it needs to assume that the ranking algorithm guarantees O(log T) sub-optimal item selections. Is O(log T) critical? Can it be O(sqrt(T))? The problem is that O(log T) seems too specific. Instead, it makes more sense to assume the ranking algorithm is no-regret, i.e., it selects suboptimal items in o(T) rounds. In that case, does Theorem 4 still hold?

**Questions:**

(1). Can the author make the assumption in Theorem 4 more prominent?

(2). Is O(log T) sub-optimal item selections a critical assumption in Theorem 4? Can the authors extend the result to arbitrary no-regret ranking algorithms that guarantee o(T) suboptimal item selections?

**Limitations:**

Yes

---

> ### Author Rebuttal · Authors · 2023-08-09
>
> **Q1**. *Can the author make the assumption in Theorem 4 more prominent?*
>
> **A1**. Thanks for the suggestion. We agree it is better to define the assumption that the ranking algorithm would choose suboptimal items sublinear times more explicitly. We will make it as a separate assumption in the updated version.
>
>
> **Q2**. *Is $O(\log T)$ sub-optimal item selections a critical assumption in Theorem 4? Can the authors extend the result to arbitrary no-regret ranking algorithms that guarantee $o(T)$ suboptimal item selections?*
>
> **A2**. Thanks for raising this excellent point. We can indeed extend Theorem 4 to arbitrary ranking algorithm with $R(T) = o(T)$ suboptimal item selections. The chosen time of the target item will become at least $T-R(T)$ in expectation, and the cumulative attack cost is at most $C(T) \le O\left(\sum_{a<L}(\Delta_a + 4\beta(1)) R(T) \right)$.
> The changes in the proof will be line 473 and Eq. (39) in the Appendix. For line 473, since Algorithm 5 still makes the expected reward of any item $l \notin {a}^*$ be worse than that of the target item in the post-attack problem, the number of target item pulls determined by the ranking algorithm directly becomes $T-R(T)$. For Eq. (39), the replacement of $\mathbb{I}\\{a_{i,t} \notin a^*\\}$ in Eq. (38) is only affected by the ranking algorithm, thus we can use $R(T)$ to replace $\log T$ in Eq. (39).
> Notice that Algorithm 5 does not have to know $R(T)$ beforehand, so it works without knowing the details of the ranking algorithms. We will include this extension in the updated version.

---

> > ### Comment · Reviewer_E7da · 2023-08-20
> >
> > Thank you for your response. My questions are well addressed in the rebuttal. After reading other reviewers' comments, I decided to keep my current score.

---

### Official Review · Reviewer_carj · 2023-07-07

**Soundness:** 3 good
**Presentation:** 3 good
**Contribution:** 3 good
**Rating:** 5
**Confidence:** 4

**Summary:**

The paper introduces adversarial attack on online learning to rank. The attacker aims to fool the online learning to rank algorithm to choose the target item. The paper studied three victim models: stochastic L-armed bandit, OLTR with position-based model and OLTR with cascade model. Authors proposed three attack algorithm respectively for three victim models and a general attack strategy to attack arbitrary OLTR algorithms without knowing the details of the algorithm. Their theoretical analysis and experimental results show that the proposed attack strategies can manipulates the learning agent into choosing the target attack item in linear time but incurring a sublinear cost.

**Strengths:**

The paper studied adversarial attacks on online learning to rank with click feedback, which as far as I know is the first work in this field. The paper provides some theoretical guarantees on their proposed algorithms. The results show that the attack strategies can successfully attack OLTR algorithms with sublinear cost. The paper is well-written and well-organized. The attack strategy on the UCB algorithm clearly shows that the differences between this work and previous work about reward attack on bandits.

**Weaknesses:**

Overall, the paper is well-written and has good technical content, but I have the following concerns:

1. An intuitive attack method is that the attacker first chooses an action (list) $a$ including the target items and sets the reward out of this list to 0. The authors name this as $Trivial_{k-1}$. The authors empirically compared the proposed methods with $Trivial_{k-1}$, which is good. However,  $Trivial_{k-1}$ seems to keep the same cost scaling on time as the proposed attack strategies, which weakens the contribution of this work.

2. There is no lower bound about the attack cost and any information. Compared with $Trivial_{k-1}$, The proposed attack algorithms improve the attack cost by a constant ratio. How far is the proposed attack strategies to the optimal?

**Questions:**

See in weakness 2.

**Limitations:**

Yes.

---

> ### Author Rebuttal · Authors · 2023-08-09
>
> **Q1**. *Trivial$\_{K-1}$ seems to keep the same cost scaling on time as the proposed attack strategies.*
>
> **A1**. We admit that the attack cost of Trivial$\_{K-1}$ is in the same order of $O(\log T)$ as our proposed algorithms.
> However, our algorithms achieve problem-dependent attack costs adaptive to the problem difficulty, which have better problem-dependent parameters than those for Trivial$\_{K-1}$ (see A3 to Reviewer i6fi).
> Trivial$\_{K-1}$ needs to pay unnecessarily large costs as it attacks all items outside the built set using the most aggressive attack values thus is not adaptive to the problem difficulty.
> Our experimental results in Figures 1 and 5 also validate our theoretical findings.
> We would like to highlight that we are the first to propose the attack algorithms with problem-dependent attack costs against OLTR with click feedback, which requires more involved fine-grained cost analyses than the analysis of the cost order in time horizon $T$.
> Such problem-dependent cost analyses are also the main focus of previous works on $L$-armed stochastic bandits [9, 10], which is essential to understanding the vulnerability of bandit algorithms under adversarial attacks.
>
> **Q2**. *There is no lower bound about the attack cost.*
>
> **A2**. The lower bound of the attack cost is hard to derive since the attacks will change the stochastic environment to a non-stochastic environment in which the behaviors of the stochastic bandit algorithms are unclear. The only known lower bound result [A] is for attacks against adversarial bandits where the pre-attack and post-attack environments are both adversarial.
> Such lower-bound analysis in the adversarial setting is a tractable question since the performance of the adversarial bandit algorithms under both the pre-attack and the post-attack environments is measurable.
> However, in the stochastic setting, this is not the case since the performance of the stochastic bandit algorithms under the post-attack adversarial environments is unclear. Hence, all previous works on attacks against stochastic bandits [9, 10, 11] do not have lower-bound results.
> Although the lower bound of the attack cost against stochastic bandits is still an open problem, our proposed algorithms for OLTR with click feedback achieve the same dependency on problem-dependent parameters $\Delta_a$ in the asymptotic costs as that for stochastic bandits with continuous and unbounded feedback [9, 10].
>
> [A] Yuzhe Ma, and Zhijin Zhou. Adversarial Attacks on Adversarial Bandits. In ICLR, 2023.

---

> > ### Comment · Reviewer_carj · 2023-08-17
> > **Thank you for the response.**
> >
> > I read the rebuttal and my opinion has not changed

---

### Official Review · Reviewer_i6fi · 2023-07-10

**Soundness:** 2 fair
**Presentation:** 3 good
**Contribution:** 2 fair
**Rating:** 5
**Confidence:** 4

**Summary:**

This paper studies adversarial attack to online learning to rank, where a successful attack to an OLTR algorithm is such that a suboptimal item is recommended for linear in T times while with log T cost. This paper approaches this attack problem by studying attack to discrete MAB as warmup and then moves on to attack UCB based OLTR algorithms under position and cascade models. Lastly, it ends up with an attack strategy for general algorithms and click models, both theoretical and simulation results are provided to justify the attacks being successful.

**Strengths:**

1. Adversarial attack to online learning to rank makes up the foundation for studying robust/trustworthy recommendation systems. I agree some research into this topic is a must-do.
2. This paper is well written and easy to read. It appears to have a thoughtful path to break down the OLTR attack problem in a few pieces and tackle it step by step, which is good. However, I doubt, which is also my major concern for this paper, that are some of the steps necessary to derive a successful attack? See details in weakness I list below.

**Weaknesses:**

**Possibly overcomplicates the problem**

There’s another work [1]https://arxiv.org/pdf/2305.19218.pdf on arxiv that seems to study the same problem as this paper. This paper retains novelty as these two can be considered as concurrent works according to the arxiv one’s submission date. But interestingly, [1] seems to have conflicted results from this paper:

The major tension lies in this paper proposes to alter clicks of unwanted arms with a sophisticated probability (e.g. in Alg.5), while [1] simply set 1 to 0 when an unwanted arm has a click. Moreover, this paper tests in simulation a baseline attack Trivial_1, which is close to the attack proposed by [1] (except for  [1] has a cutoff time for attack), and this paper claims Trivial_1 fails to attack according to the plot. And not surprisingly, theorem and simulation results in [1] supports their attack.

So this finding casts some doubt on this paper whether it overcomplicates the problem, as the attack proposed in [1] is way simpler with no need to compute a sophisticated probability to attack. Here are some clues in this paper that I think questionable:

1. In Section 3.2, it claims that the challenge in applying previous MAB attack to discret MAB is $\alpha_t$ in (1) can be invalid (>=1) when the altered reward should keep binary, which leads to all the effort in finding a new  $\alpha_t$ with care. But simply setting all returns for unwanted arms to be 0 may also get the job done.
2. Simulation section shows that Trivial_1 fails while Trivial_{k-1} succeeds,I don’t think Trivial_{k-1} has essential difference than Trivial_1 as in the special case k=2 they are identical. So maybe there’s something going wrong with the plots or interpretation.

**Questions:**

Major question please refer to weakness 1.

**Limitations:**

Identified Limitations are listed in Weakness and Question.

---

> ### Author Rebuttal · Authors · 2023-08-09
>
> **Q1**. *Comparison with the other work [1].*
>
> **A1**. We thank the reviewer for pointing us to this related work. We kindly remind that the submission date of this arXiv paper is after the NeurIPS deadline. We would like to make some clarifications on the difference between our work and theirs.
>
> 1) Attack objective. Our attack objective is to manipulate the learning agent to put the target item into the recommended list, while their goal is to place the target item on top of the list. These objectives are related but different: their objective requires additional assumptions about the candidate items and the prior knowledge available to the attacker, while our model does not make any assumption about the items and our algorithms do not need any prior knowledge.
>
> 2) Algorithm and assumption. The click poisoning attack strategy attack-then-quit (ATQ) in [1] can only be applied to elimination-based rankers BatchRank and TopRank: it not only sets the clicks of the non-target items to be zero but also sets the clicks of the target item to be one, thus will incur linear attack costs for any non-elimination-based ranker (e.g., PBM-UCB, CascadeUCB) without a cutoff time. Our attack algorithms are more relevant to their generalized list poisoning attack strategy GA against PBM-UCB and CascadeUCB. For their GA algorithm, they assume that there exist $2K-1$ low attractiveness items, and the algorithm needs to know them in advance; this is due to their objective of placing the target item on top of the list. However, with our objective of placing the target item in the list, we do not make any assumption about the items. Also, their theoretical analysis of GA mainly holds for rankers with a unique optimal list (from their Definition 2), while our Algorithm 5 can successfully attack arbitrary OLTR rankers.
>
> 3) Attack cost. We focus on fine-grained attack design against OLTR with click feedback and provide problem-dependent attack costs adaptive to the difficulty of the problem ($\Delta_a$ in Table 1), while they mainly consider the order of the attack cost in $T$ to be sublinear, i.e., $o(T)$. Such problem-dependent cost analyses are also highlighted in previous works [9, 10].
>
> To sum up, with our objective of manipulating the learning agent into choosing the target item, we propose attack algorithms without additional assumptions and prior knowledge about the candidate items. Our attack design and analyses are more involved as our goal is to derive fine-grained problem-dependent attack costs adaptive to the problem difficulty, which cannot be achieved by simple attack design and play an essential role in understanding the vulnerability of OLTR under adversarial attacks.
>
> **Q2**. *Conflicted results about the Trivial$_1$ baseline.*
>
> **A2**. As mentioned in A1 (2), the GA algorithm in [1] uses the known low attractiveness items to build a list with the target item, and then sets the clicks of all items outside the list to be zero. Our Trivial$_1$ is actually different from GA since it does not build any list and just sets the clicks of all items except the target item to be zero. Trivial$_1$ fails because it always needs to attack the other clicked items in the recommended list and incurs linear attack costs (see A2 to Reviewer 7TEH for more details). Thus, our experimental results do not conflict with those in [1].
>
> **Q3**. *Why not simply setting all returns for unwanted arms to be 0?*
>
> **A3**. As discussed in A1 (3), we would like to design attack algorithms with fine-grained problem-dependent attack costs adaptive to the problem difficulty, since non-adaptive attack strategies need to pay unnecessarily large attack costs. Notice that the problem-dependent cost analysis would be much more challenging than the analysis of the cost order in time horizon $T$, which is one of our technical contributions.
> For the $L$-armed bandit problem with binary feedback, simply setting all returns of non-target arms to be $0$ can indeed succeed, but its attack cost will depend on $\mu_a$ instead of $\Delta_a$, which is undesirable compared to previous results in [9, 10]. For OLTR with click feedback, Trivial$\_{K-1}$ has the same issue that the problem-dependent parameters in the attack cost can be much worse than those of our proposed algorithms. This is validated by our experiments on both synthetic and real data: Figures 1 and 5 show that our algorithm pays around $40\\%$ less costs than Trivial$\_{K-1}$ for both PBM-UCB and CascadeUCB.
>
> **Q4**. *Trivial$\_1$ and Trivial$\_{K-1}$ when $K=2$.*
>
> **A4**. Trivial$\_{K-1}$ means taking $K-1$ items out to build a set with the target item, so when $K=2$, it will still take $1$ item out to build a set with the target item and is different from our Trivial$\_{1}$ that does not build any set. As mentioned in Q2, Trivial$\_1$ needs to attack all clicked non-target items in the recommended list, while Trivial$\_{K-1}$ only attacks the non-target items outside the built set. The performance of Trivial$\_{K-1}$ is still better than Trivial$\_{1}$ when $K=2$. Thanks for capturing this naming issue and we will change these names to Trivial$\_{\text{all}}$ and Trivial$\_{\text{set}}$ and in the updated version.

---

> > ### Comment · Reviewer_i6fi · 2023-08-22
> >
> > Thanks for your response, most of my questions have been addressed. I'd like to raise my score by 1.

---

### Official Review · Reviewer_7TEH · 2023-07-27

**Soundness:** 4 excellent
**Presentation:** 4 excellent
**Contribution:** 3 good
**Rating:** 5
**Confidence:** 3

**Summary:**

This paper studies attack strategies against UCB-based OLTR algorithms for position-based click model and cascade click model. Compared with prior works in adversarial attacks, OLTR problem leads to the additional challenge that the rewards are discrete and bounded, which requires different attack method design, i.e., to ensure the post-attack reward is valid. The authors further propose attack strategy that works for OLTR algorithms with general click models, at the expense of additional attack cost compared with the strategies designed for specific click models.

**Strengths:**

This paper is well-written and seems to be technically sound.

To the best of my knowledge, this is the first work that provides rigorous solutions and analysis for attacking OLTR algorithms.

The designed attack strategy for discrete and bounded reward might be of independent interest, e.g., it may be applied to other problem settings with discrete reward.

**Weaknesses:**

I think a more intuitive (and self-contained) explanation, on why the use of conservative estimation can address the challenge caused by bounded reward, would be helpful. The current writing of Section 3.2 requires some knowledge of of [9] to fully understand the intuition and appreciate its novelty.

**Questions:**

Can the authors provide more explanation on why constructing the set $a^{\star}$ by randomly selecting $K-1$ items, in addition to the target item $L$? Since the goal is to make target item $L$ appear in the recommended set $a_{t}$, why do we want to spend cost on making the learner recommend the $K-1$ randomly sampled items as well? Is it possible to simply making sure L replaces the last item in $a_{t}$?

The reason the current design (for single target item) seems to be sub-optimal is that, this algorithm can be directly applied to the case where there are more than 1 but less than K+1 target items, without incurring additional cost, i.e., the algorithm will need to spend cost in making the learner pull $K-1$ randomly selected items anyway.

---

> ### Author Rebuttal · Authors · 2023-08-09
>
> **Q1**. *Intuitive explanation of conservative estimation.*
>
> **A1**. The main idea of the attack algorithm in [9] is to calculate the attack value per round such that the post-attack empirical estimates of the non-target arms are always less than that of the target arm (i.e., Eq. (1)). However, with binary feedback, if we follow their original design, the calculated attack value can be greater than one, thus it is an invalid attack value and Eq. (1) required by their theoretical analysis can not hold.
> We resolve this issue by using the conservative estimation of the target arm: a conservative lower bound of the target arm's mean is used in Eq. (2) and Eq. (3) to determine the attack value. Intuitively, it makes the attacker more conservative to use a small attack value (that may result in an invalid large attack value in later rounds). It ensures the calculated attack value is always valid and Eq. (4) holds for all rounds (required by our new theoretical analysis). We will revise the explanation of the conservative estimation in the updated version.
>
>
> **Q2**. *Explanation on why constructing the set by randomly selecting items. Is it possible to simply making sure $L$ replaces the last item in ${a}_t$?*
>
> **A2**. Different from the simple $L$-armed bandit setting where only a single item will be checked by the player, in OLTR, all items in the recommended list can be checked by the player (depending on the click model). Thus, in addition to putting the target item in the recommended list, we also need to ensure that the attacker does not have to attack the other items in the list too often. By randomly selecting $K-1$ items to build a set with the target item, our attack algorithm will only attack those items outside the set, and the recommended list will gradually only contain the items in the set. This is a simple solution to control the attack costs on the other items in the recommended list and we prove in Theorems 2, 3, 4 that it only requires sublinear costs.
>
> **Q3**. *The current design (for single target item) seems to be sub-optimal.*
>
> **A3**. For the single target item case, if we know the best $K-1$ items in advance, a better solution might be attacking the remaining items except for the target item (i.e., using the best $K-1$ items to build the set with the target item). However, we consider the setting without prior knowledge of all items; learning the best $K-1$ items while doing attacks can be complicated since the attacker's observation depends on the player's action and the player's action is affected by the attacks. Although a better solution might be possible with a more carefully design of the joint learning and attacking, our idea of randomly sampling $K-1$ items can be directly applied in attacking most OLTR algorithms with different click models, while learning the best $K-1$ items needs to be algorithm/model-specific. We will add this discussion in the updated version.

---

> > ### Comment · Reviewer_7TEH · 2023-08-19
> > **Thank you for the response**
> >
> > The response from authors is very helpful. I have also read the comments from other reviewers, as well as responses from authors.
> > Now I have a better understanding of the algorithm design and its guarantees.
> >
> > From authors' response to Reviewer i6fi, the main advantage/contribution of the proposed attack algorithm is that its attack cost has better dependence on problem parameter $\Delta_{a}$, since it can conduct finer-grained attack (compared with setting reward of non-targeted item to 0) using the additional knowledge about the victim algorithm and the click model.
> >
> > However, what I feel is still unsatisfactory and lacks justification is proposed algorithm's dependence on $K$. As I mentioned in my original comments, since the goal of this paper is only to make sure the target arm appears in the recommended list (which is weaker than the goal of the work mentioned by Reviewer i6fi, that needs to put the target arm on top of the list), the current algorithm's design appears to be sub-optimal on this part. In this setting, joint learning and attacking might be needed to avoid spending unnecessary costs on the top $K-1$ arms.
> >
> > The authors response to these two questions seems to be a bit conflicting.
> > For the dependence on $\Delta_{a}$, the authors argued their algorithm is "better" than the work mentioned by Reviewer i6fi, since it can utilize algorithm/model-specific information to conduct finer-grained attack, while for the dependence on "K", the authors argued their algorithm is also "better" than the suggested joint learning and attacking design, because it does not need algorithm/model-specific information.
> >
> > Therefore, I decided to lower my score.

---

> > > ### Author Response · Authors · 2023-08-20
> > > **Thank you for the follow-up**
> > >
> > > We appreciate the reviewer’s follow-up comments.
> > >
> > > Regarding the advantages of our proposed attack algorithms, please note that our Algorithms 2 and 4 are specifically designed for attacking PBM-UCB and CascadeUCB, while our Algorithm 5 can indeed successfully attack arbitrary OLTR algorithms without knowing the algorithm/model-specific information. Although Algorithms 2 and 4 have better problem-dependent cost bounds than that of Algorithm 5 on their specific target algorithms, all of them have better problem-dependent parameters ($\sum_{a}\Delta_a$) than those of the trivial attack ($\sum_{a} 1 / \Delta^2_{\min}$ in [1]) owing to our fine-grained attack value design that does not fully rely on the algorithm-specific information.
> > >
> > > Regarding the algorithm’s dependence on the recommendation list size $K$, we would like to clarify that the attack costs always have the linear dependence on $L-K$, as we can only keep $K$ items (including the target item) unchanged in the post-attack problem. Only the dependence on $\Delta_a, \forall a\notin a^*$ could be suboptimal for our chosen unchanged set $a^*$. However, since $\Delta_a$ is usually small ($\Delta_a \le 0.15$  in the experiments from [4, 5]), we would expect the cost gap also to be small (the maximum gap is $\sum_{a \in [K-1]} \Delta_a  - \sum_{a \in a^*} \Delta_a $) and our algorithms just need to pay slightly higher attack costs than that in the ideal case of knowing the best $K-1$ items. We agree with the reviewer that there may exist a different attack strategy with better dependence on $\Delta_a, \forall a\notin a^*$, but as discussed in A3, it requires algorithm-specific design for learning the best $K-1$ arms while making attacks. Our idea of random sampling is a simple but effective solution that can be applied to attacking arbitrary OLTR algorithms.
> > >
> > > Sorry for the confusion regarding the generality of random sampling. We are meant to say that our idea of randomly sampling $K-1$ items itself does not require algorithm/model-specific information. In contrast, learning to find the best $K-1$ while attacking requires more intricate design tailored to specific OLTR algorithms. This random sampling idea also needs to be incorporated with additional fine-grained attack design to achieve the desired attack cost guarantees. For example, both Algorithm 2 and 4 utilize such random sampling and they are tailored to attacking PBM-UCB and CascadeUCB; Algorithm 5 also uses random sampling but it does not require any algorithm/model-specific information.
> > >
> > > Thank you again for your valuable time and insightful feedback. We are more than happy to answer any follow-up questions.

---

### Author Response · Authors · 2023-08-17
**A gentle reminder to reviewers**

Dear reviewers,

Thank you again for your valuable comments and suggestions. Since we are approaching the end of the discussion phase, we would appreciate it if you could read our responses and let us know if you have any follow-up questions.

Best regards,
Paper6757 Authors

---

### Decision · Program_Chairs · 2023-09-21

**Decision:**

Accept (poster)

**Comment:**

This paper studies reward poisoning attacks against online learning to rank (OLTR) where the rewards are discrete and bounded.  Attack strategies are proposed against UCB-based algorithms for position-based click model and cascade click model, and both achieve T-O(log T) target item selections with only O(log T) attack cost. The paper also designed a general attack strategy for arbitrary click models that achieve O(log T) sub-optimal item selections.

Overall, the reviewers are all positive to the paper.  The reviewers appreciate that the problem is well-motivated and the theoretical results are sound. There are concerns regarding the attack target and advantages against baselines, and the authors' responses resolved the concerns. I am happy to recommend acceptance. The authors are suggested to incorporate the response and discussion into the final version.